# Sodium molybdate does not inhibit sulfate-reducing bacteria but increases shell growth in the Pacific oyster *Magallana gigas*

**Roxanne M. W. Banker** [1¤*], **Jacob Lipovac**[1], **John J. Stachowicz**[2], **David A. Gold**[1*]

**1** Department of Earth and Planetary Sciences, University of California, Davis, California, United States of America, **2** Department of Evolution and Ecology, University of California, Davis, California, United States of America

¤ Current address: Department of Invertebrate Zoology and Geology, California Academy of Sciences, San Francisco, California, United States of America

* rbanker@ucdavis.edu (RMWB); dgold@ucdavis.edu (DAG)

## Abstract

Recent work on microbe-host interactions has revealed an important nexus between the environment, microbiome, and host fitness. Marine invertebrates that build carbonate skeletons are of particular interest in this regard because of predicted effects of ocean acidification on calcified organisms, and the potential of microbes to buffer these impacts. Here we investigate the role of sulfate-reducing bacteria, a group well known to affect carbonate chemistry, in Pacific oyster (*Magallana gigas*) shell formation. We reared oyster larvae to 51 days post fertilization and exposed organisms to control and sodium molybdate conditions, the latter of which is thought to inhibit bacterial sulfate reduction. Contrary to expectations, we found that sodium molybdate did not uniformly inhibit sulfate-reducing bacteria in oysters, and oysters exposed to molybdate grew larger shells over the experimental period. Additionally, we show that microbiome composition, host gene expression, and shell size were distinct between treatments earlier in ontogeny, but became more similar by the end of the experiment. Although additional testing is required to fully elucidate the mechanisms, our work provides preliminary evidence that *M. gigas* is capable of regulating microbiome dysbiosis caused by environmental perturbations, which is reflected in shell development.

## Introduction

Disentangling the relationship between environment, microbiome, and host fitness is essential for predicting how marine taxa will respond to climate change. Marine invertebrates that build calcified skeletons are under particular scrutiny, given how climate change driven ocean acidification is predicted to make skeleton growth more difficult [1]. Both the host and its microbiome (the holobiont) are likely to play an important role in the formation of such skeletons [2,3] as most microbiomes are species specific and correlate strongly with characteristics of the hosts' innate immunity and, to a lesser extent, host trophic habits and local environmental conditions [4]. The relationship between marine invertebrates and their resident microbiota

**Data Availability Statement:** The raw data from the genetic analyses have been submitted to NCBI under BioProject PRJNA762081. The code as well as intermediate files are preserved on GitHub at

https://github.com/DavidGoldLab/2021_
Magallana_Sodium_Molybdate_RNAseq and
https://github.com/Roxanne-Banker/Oyster-SM.
Images of shells are available through Harvard
Dataverse using the link https://doi.org/10.7910/
DVN/C9CSB3.

**Funding:** "The role of microbial sulfate reduction in
oyster biomineralization: implications for changing
oceans." PIs: D. A. Gold & J. J. Stachowicz.
Postdoctoral researcher: R. Banker. UC Davis
Microbiome Special Research Program. The
funders had no role in study design, data collection
and analysis, decision to publish, or preparation of
the manuscript.

**Competing interests:** The authors have declared
that no competing interests exist.

affects host health, growth, mineralization, and resilience to stress, all of which impacts the final form of a calcified skeleton [5–7].

Sulfate-reducing bacteria are of particular interest regarding their role in animal calcification because of their ability to affect carbonate dynamics via metabolic activity [3,8–13]. Sulfate-reducing bacteria in ocean sediments can precipitate calcium carbonate under oxic and anoxic conditions, and may be responsible for more than half of the carbon oxidized through the seafloor [8,14–18]. They have additionally been implicated in the precipitation of calcium carbonate in living organisms as diverse as stromatolites, rhodoliths, and agglutinated polychaete worms [19–21]. Sulfate reducing bacteria are considered the most likely candidates for aiding in the shell growth of molluscs such as oysters, potentially colonizing the extrapallial fluid between the shell and mantle and/or fluid-filled spaces within the shell itself [22]. Oysters also undergo periods of anoxia when their valves are shut [23], which potentially provides additional opportunities for anaerobic sulfate reducers. Still the degree to which sulfate reducers drive carbonate precipitation, particularly in animals, remains controversial.

To test of the role of sulfate reducers in shell formation, we reared larvae of Pacific oyster (*Magallana gigas*, formerly *Crassostrea gigas*) through settlement while exposing them to pulses of sodium molybdate, a compound that has been used in previous experiments to inhibit sulfate-reducing bacteria in marine systems [24,25]. A previous study showed that the abundance of sulfate-reducing bacteria in juvenile oysters is not correlated with enhanced chalk expression, but is correlated with denser shells [25]. Here, we build on this work to determine how exposure to this compound affects oyster microbial communities, gene expression, and shell growth at early post-settlement stages, when mortality can be high and rapid growth and shell extension is correlated with long-term fitness [26].

## Materials and methods

### Study organism and broodstock

On July 30, 2019, broodstock oysters were obtained from the Hog Island Oyster Company. Upon arriving at the Bodega Marine Laboratory, all oysters were soaked in a 60 ppm solution of sodium hypochlorite (bleach) for 1-hour according to regulations from the California Department of Fish and Wildlife. Oysters were left in air for ~1.5 hours before being placed in a 1500L tank of filtered seawater, at approximately 18˚C. Broodstock were fed daily with 5 gallons of algal culture composed of either *Isochrysis sp.* (strain CCMP463, Bigelow National Center for Marine Algae and Microbiota) or *Nannochloropsis sp.* (strain CCMP525, Bigelow National Center for Marine Algae and Microbiota). Water was flushed every 2–3 days by inserting a standing pipe into the drain at the bottom of the tank and allowing incoming water to displace tank water. This was done for 30 minutes for each flush. A full water change, including a complete draining and cleaning of the tank, occurred approximately every 7 days.

The visceral mass was removed from the shell and placed in a weigh-boat. Female eggs were stripped by gently rinsing seawater over the scored gonad. Eggs were then passed through a 75 µm sieve to exclude errant tissue. Finally, the eggs were rinsed into a bucket of seawater. Four females were stripped in this way into different buckets, and eggs were allowed to incubate in seawater for 45–60 minutes to condition.

After we prepared four buckets of eggs for conditioning, male sperm were stripped in a similar manner. Sperm was passed through a 20 µm sieve in order to exclude tissue. Harvested sperm was diluted into 500 mL of filtered seawater (FSW), then aliquots were added to eggs until there were ~2–4 sperm cells surrounding eggs in sample aliquots viewed under the microscope. Embryos from two crosses, totaling approximately 16,000,000 individuals, were selected based on fertilized egg quality. Fertilized eggs were considered good quality if they

had a regular, round shape with a plump appearance, as opposed to ovate or wrinkly. The number of embryos was quantified by pipetting measured volumes of the embryo solution onto a welled microscope slide. Embryos in each droplet were counted under a microscope, and counts were averaged across wells. We extrapolated embryo density based on these counts. Embryos were placed into a 1500 L tank with filtered seawater at ~23˚C aerated by an air stone. The larval tank was supplied two gallons of *Isochrysis sp*. algae, and allowed to incubate for 48 hours, which is when most individuals examined under a microscope had reached the "D-hinge" developmental stage. After 48 hours, the larvae were separated evenly into treatment buckets. This was done by siphoning water from the top of the tank and onto a 35 um sieve, thus isolating larvae on the screen. Larvae were then rinsed into a bucket where larval densities could be estimated using the same method applied to estimate embryo densities. Finally, the larvae were separated evenly into 9 buckets (three treatments, three bucket replicates each) at a density of ~10 larvae/mL.

## Experimental design and treatment conditions

Larvae were raised under two experimental treatments: control seawater (pH≈8.00) and seawater with sodium molybdate added (Fig 1). As a functional analog of the sulfate ion, molybdate can be transported into bacterial cells during cellular respiration, which deprives the microbe of sulfur [27,28], thus acting as a sulfate-reduction inhibitor. Each treatment was maintained in three 10-gallon food-grade buckets, which were pretreated for 3 days, with the water changed each day. Filtered seawater from the Bodega Marine Lab intake system was pretreated with air bubbled through an airstone. Sodium molybdate was added to the appropriate treatment buckets at a concentration of 12.8 mg/L after each water change (i.e. once every three days). Each bucket received ambient air bubbled into buckets through an airstone to maintain oxygen. Crushed oyster shell, that was size sorted to 175 um and autoclaved, was added to buckets on September 16 (20 days post-fertilization) to act as a substrate for

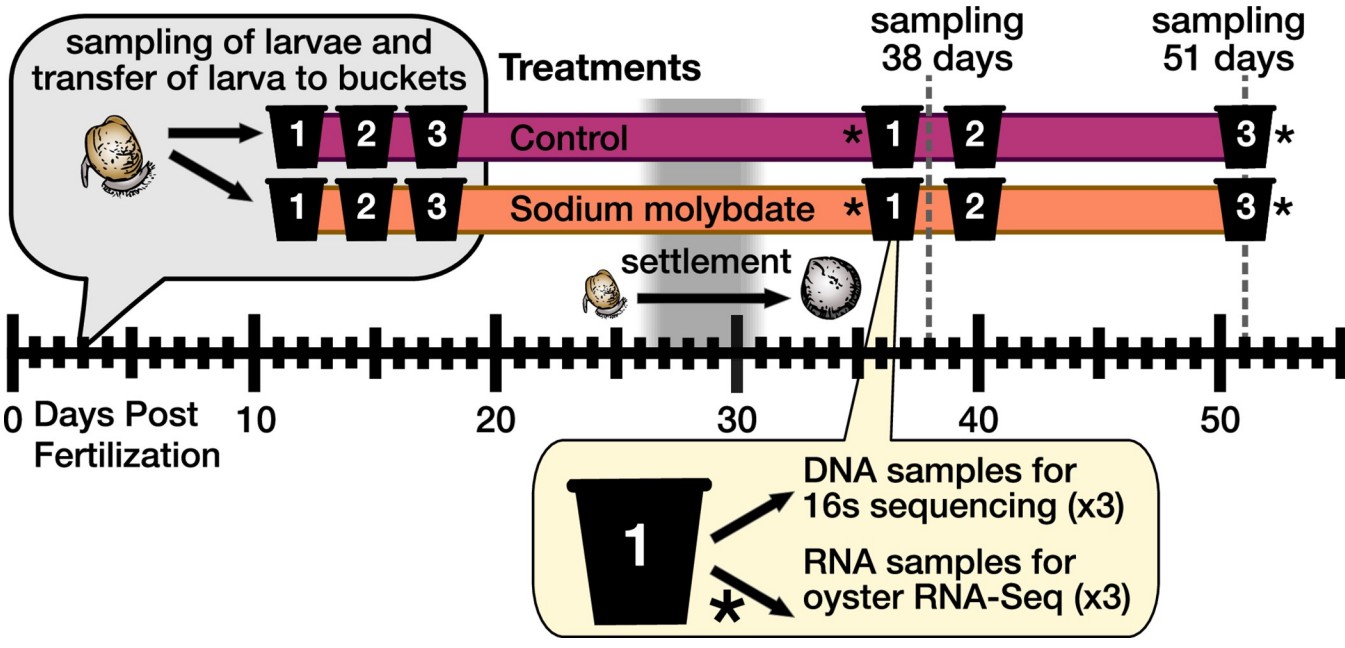

**Fig 1. Schematic of experimental design.**

settlement (S1 Table). Observations of buckets revealed that settlement for individuals began on day 26 and the remaining organisms had settled by day 31 post-fertilization (S1 Table).

Despite adding crushed shells for settlement, the oysters settled exclusively on the sides and bottom of buckets. Juvenile oysters were collected by scraping a pea-sized volume of settled juvenile oysters off of the sides of buckets using a small spatula. Oyster samples were quickly rinsed in ethanol to remove any external microbes before being frozen at -80˚C. On Day 38 post-fertilization, two of the three replicate buckets from each treatment were destructively sampled to collect juvenile oysters for nucleotide extraction. Naming of the samples follows the structure "Treatment-Time-Bucket-Replicate"; the three replicates from control bucket 1 are, for example, hereafter referred to as "C-38D-1-[1–3]", while the three replicates from bucket 2 of the sodium molybdate treatment are "SM-38D-2-[1–3]". Our original goal was to grow juvenile oysters out to ~38 days post settlement, and we decided to maintain three repli-cate buckets for each treatment in case a mortality event affected our ability to collect samples. When all buckets made it past settlement without issue, we culled two of the three buckets from each treatment for data collection, but allowed the third bucket to grow out to 51 days post-settlement. This was done so that we could also collect additional data on how the Pacific oyster microbiome and transcriptome change during early post-settlement development, although it limited the types of analyses we could perform on the 51-day data.

**Oyster diet.**   After D-hinge veliger larvae were transferred to experimental buckets from the group culture, animals were fed a diet of *Isochrysis sp*. at ~30,000 cells/mL until September 4th (8 days post-fertilization). Thereafter, feeding concentrations were gradually increased according to established larval care protocols employed by the Aquatic Resources Group at the Bodega Marine Lab. Algal concentrations were increased to ~100,000 cells/mL by September 19 (23 days post-fertilization). When settlement was first observed in buckets on September 22 (26 days post-fertilization), feeding was increased to *ad libitum* to encourage rapid growth.

**Static culture water changes.**   In order to maintain culture quality, water in buckets was changed every three days. For the first water change that took place on September 1 (5 days post-fertilization) a 35 μm sieve was used to isolate larvae. Progressively larger sieves were used over the course of the experiment (S1 Table).

**Water chemistry monitoring.**   The following parameters were measured and recorded twice daily for all culture buckets: pH, temperature, dissolved oxygen (DO), and salinity. This was done using a Pinpoint® pH controller and probe, a Sper Scientific Dissolved Oxygen Meter Kit (which measured both DO and temperature), and a refractometer, respectively. In addition, old oyster culture water was sampled during each water change. Water samples were also taken for incoming seawater (water that was used to restock larvae after a water change) starting on the water change that occurred on September 7, 2019. These samples to be analyzed spectrophotometrically for pH (total scale) were collected in 125 mL glass bottles with a posi-tive meniscus, and were spiked with mercuric chloride. Caps were wrapped with parafilm, and bottles stored at 4˚C until analyzed on an Ocean Optics Jaz Spectrophotometer EL200 (SD +/- 0.003) using *m*-cresol purple (Dickson et al., 2007). A calibration regression was produced for each batch of dye (*m*-cresol) and calibrated against Tris for a <0.1 pH offset.

Samples for total alkalinity were collected in 250 mL Nalgene© tubes with a small amount of headspace, and immediately frozen at -20˚C. These were run on an automated Gran titra-tion on a Metrohm 809 Titrando (SD +/- 4.2 umol/kg); acid concentrations standardized using Dickson certified reference materials.

Mean alkalinity was calculated for each bucket on the basis of samples that were successfully analyzed for that bucket, though the number of samples that was used for each alkalinity bucket mean was variable (S2 Table). Mean total alkalinity values for each bucket were assigned to each spectrophotomically derived pH value that also belonged to the same bucket.

These values were used as a final correction to pH measurements, using CO2calc (Robbins et al., 2010) with $pK_1$ and $pK_2$ CO$_2$ equilibrium constants from Millero (2010) and KHSO$_4$ from Dickson (1990).

To confirm that the experimental design was effective in regards to maintaining the same pH conditions in both treatments, a Student's t-test was used to compare mean pH between treatment groups. This procedure was performed with pH values measured on the Ocean Optics Jaz Spectrophotometer and corrected with mean bucket total alkalinity measurements. All statistical tests were performed in R (version 3.5.3).

## 16S rRNA sample preparation and sequencing

ZymoBIOMICS DNA Miniprep Kits were used to extract total DNA from larval (1000s individuals) and juvenile (100s of individuals) samples. Total DNA was sent to the Integrated Microbiome Resource (IMR; https://imr.bio/) at Dalhousie University (Halifax, Nova Scotia), for PCR amplification and sequencing. Sequencing libraries were prepared by targeting the V4-V5 regions of the 16S rRNA gene for amplification using the 515F (`GTGY-CAGCMGCCGCGGTAA`) and 926R (`CCGYCAATTYMTTTRAGTTT`) primer set. Sequencing was performed on the Illumina HiSeq platform.

## Microbiome analysis and visualization

The resulting fastq files were analyzed in R (v. 3.6.2) using dada2 (v. 1.14.0), phyloseq (v. 1.30.0), vegan (v. 2.5.6), FSA (v. 0.8.27), the SILVA taxonomic training dataset (v. 132), and ggplot2 (v. 3.2.1) [29–39]. A detailed walkthrough of subsequent analyses in R can be found in the R-markdown summary file hosted on github (https://github.com/Roxanne-Banker/Oyster-SM).

During the filtering step in dada2 (filterAndTrim), reads were truncated to reduce estimated error rates and optimize sequence merging in subsequent steps. We chose not to correct for 16s rRNA gene copy number because the proper approach remains an open question and tools developed for this purpose have shown to be unreliable thus far [40,41]. The DADA2 sample inference algorithm (using the dada command) was applied to denoise forward and reverse reads, which were then merged using mergePairs. A table of amplicon sequence variants (ASVs) was generated using merged reads, chimeric sequences were removed using removeBimeraDenovo, and taxonomy was assigned using version 132 of the SILVA taxonomic training dataset formatted for DADA2 [31,33,35].

The resulting ASV and taxonomy tables were passed to phyloseq as a phyloseq object. Decontam's prevalence method was used to identify contaminants in negative control samples using a threshold of 0.1 [39]. This threshold revealed 1 possible contaminant sequence, which was removed from the dataset. Chloroplast, mitochondria, and animal sequences were also removed from the dataset. After these initial bioinformatics steps two samples, one each from Buckets SM-38D-1 and SM-38D-2 had very low ASV abundances (<20 ASVs; S3 Table), making them impossible to include in subsequent analyses, but each remaining sample had >25,000 ASVs.

Alpha diversity (i.e. within sample diversity) was characterized by calculating the number of Observed ASVs and a Shannon diversity index, which are measures of taxon richness and diversity, respectively. This was done using the *estimate_richness* function in phyloseq. The Kruskal-Wallis test was used to compare alpha diversity amongst all bucket replicates from all treatments. A linear mixed model was not used to assess bucket effects on alpha diversity because there was insufficient replication to perform the analysis.

Sample counts were transformed using varianceStabilizingTransformation in DESeq2 [42], and negative values from the log-transformation in the ASV table were replaced with zeroes to enable ordination. Sequence counts were transformed into relative abundances, then Bray-Curtis dissimilarity was calculated and the ordinate function was applied to relative abundance data in phyloseq using the Non-Metric Multidimensional Scaling (NMDS) method.

A permutational analysis of variance (PERMANOVA) was used to assess between group differences (i.e. beta diversity) by testing if mean centroids of sample categories (i.e. treatment groups) were significantly different [43]. The PERMANOVA was performed on Bray-Curtis distances calculated between samples with 9,999 permutations to account for multiple tests using the adonis function in vegan. While the PERMANOVA is more robust than other multivariate tests [44], results can be affected by variation in centroid dispersion when applied to distance-based matrices (e.g. Bray-Curtis) [45]. Differences in mean dispersions between sample categories was assessed using the betadisper function in vegan (with 999 permutations using permutest), which is a multivariate form of Levene's test of homogeneity of variances. P-values from pairwise PERMANOVA and Levene's tests were adjusted to maintain α = 0.05 using the Bonferroni method as implemented in the p.adjust function in R. This test was only applied to samples collected 38 days post-fertilization because there were not enough samples collected 51 days post-fertilization to complete this test.

To determine which bacterial taxa were driving differences in beta diversity between treatments, ASVs were collapsed to the Order level using the tax_glom function in phyloseq. Bacterial Orders were used to reduce the number of taxonomic groups for comparison while retaining the potential for being metabolically informative [46,47]. The relative abundance of Orders were compared between treatment groups at 38 days and 51 days post-fertilization using the Kruskal-Wallis test at a critical threshold of 0.05. When the Kruskal-Wallis test was rejected, the Dunn test was applied as a post-hoc, and Bonferroni corrected p-values were used to assess which pairwise comparisons for a given family were significant (α = 0.05) (S4–S6 Tables).

## Oyster RNA extraction, sequencing, and bioinformatics

Total RNA was isolated from both larval and juvenile oyster samples by adding 0.5 mL TRIZOL Reagent (Life Technologies) to sample tubes. A Pellet Pestle® Motor (Kontes) was used to grind samples. These were left to incubate for 5 minutes before 0.10 mL chloroform was added, followed by a second incubation for 2 minutes at room temperature. Samples were then centrifuged at 12,000xg at 4˚C for 15 minutes. The top aqueous layer was removed, and an equal volume of isopropyl alcohol and 1 ug of GlycoBlue (ThermoFisher, cat # AM9516) were added; this mixture was incubated for 10 minutes at room temperature, then centrifuged at 12,000xg at 4˚C for 15 minutes. The resulting RNA precipitate was washed with 70% ethanol and then dried before being dissolved in RNase free water. Concentration and contamination of extracted RNA was assessed using a NanoDrop (NanoDrop Technologies, Inc.). Samples were then submitted to the UC Davis DNA Technologies & Expression Analysis Core Laboratory, where preparation of cDNA preparation and sequencing occurred for Tag-Seq analysis. The samples were processed using the QuantSeq 3' mRNA-Seq Library Prep Kit FWD for Illumina (Lexogen), and then sequenced on an Illumina HiSeq. Due to budget limitations, we only sequenced the one of the two "38 Day" buckets that produced the highest-quality RNA extractions.

The RNA-Seq data was returned as fastq files, which were trimmed and cleaned using the BBDuk program in BBTools [48]. The cleaned fastq files were mapped against the *M*. *gigas* genome (assembly "cgigas_uk_roslin_v1"; NCBI Assembly: GCA_902806645.1) using STAR

aligner [49]. We used the Cufflinks package [50] to create a revised set of gene annotations, merging them with the original gene models. Trinotate was used to annotate the gene models [51], and PtR program packaged with Trinity was used to generate PCA plots and correlation matrices for the data [52]. We used the cuffdiff function in Cufflinks to determine differential expression, and the go-seq function packaged in Trinity to search for enriched gene ontologies in the differentially expressed genes [53].

### Scanning Electron Microscopy (SEM) and shell growth analysis

While sampling oysters for genetic material, additional individuals were carefully removed from their settlement position with a paint brush. These individuals were stored in ethanol in order to preserve shell material. Whole valves, 10 per treatment, were mounted onto an aluminum stub with carbon tape and examined using Hitachi TM300 scanning electron microscope (SEM) in the UC Davis Department of Earth and Planetary Sciences. This allowed for clear visualization of the boundary between the prodissoconch II (larval shell) and dissoconch (juvenile shell). Image J was used to measure the whole shell area (prodissoconch II and dissoconch) of the shells of specimens. To obtain an additional metric of shell growth, ten shells from each bucket, different than those mounted for SEM analysis, were weighed using a Sartorius Pro 11 digital scale.

A linear mixed effects model was fit using restricted maximum likelihood in the lme4 package in R [32,36]. The model predicted shell area and mass, respectively, with treatment as a fixed-effect parameter and bucket ID as a nested random-effect within treatment. The call for this model in lme4 was: area (or mass) ~ treatment + (1 | bucket.id/treatment). Unfortunately the small number of buckets (N = 2) per treatment made it difficult to separate the effect of bucket and treatment. These tests were only applied to samples collected 38 days post-fertilization because there were no bucket replicates collected 51 days post-fertilization. Student's t-test was applied to compare shell metrics at 51 days between treatments.

Average percent change of oyster shell area and mass between the control and molybdate treatments was calculated by first taking the mean shell area of 38 day old shells from both treatments. The percent difference was then calculated as $((\bar{X}_{SM38} - \bar{X}_{C38})/\bar{X}_{C38} * 100$, where $\bar{X}_{SM38}$ is the mean shell area (mm$^2$) for 38 day old molybdate oysters and $\bar{X}_{C38}$ is the mean shell area (mm$^2$) for 38 day old control oysters. This calculation was also used to assess percent area and mass change for 51 day old oysters.

Daily log mean growth rate was calculated for individual oysters in each bucket. Mean shell area was calculated for the larval (prodissoconch II) and juvenile (dissoconch) shell areas for oysters (n = 10) in each bucket. Mean larval shell area was divided by 31 days, the approximate age of larvae at the time of settlement to obtain an approximate daily average larval growth rate for each bucket. For oysters that were harvested at 38 days, daily mean juvenile shell growth rate was calculated by dividing mean juvenile shell area by 7 days, the amount of time elapsed between settlement and collection. Oysters collected at 51 days had mean juvenile shell area divided by 20 days (time elapsed between settlement and collection) to obtain an estimate of daily mean juvenile growth rate.

## Results

### Oyster growth

Oysters exposed to sodium molybdate exhibited greater shell growth than the control. Salinity, temperature, and pH of seawater did not differ between the two treatments (T-test, p>0.05; Table 1). After 38 days, oysters in the molybdate treatment group were on average ~186%

**Table 1. Mean ± SD of water parameters and carbonate system in culture buckets.**

|  | Measured | | | |
|---|---|---|---|---|
|  | **pH** | **Temperature** | **Salinity** | **DO** |
| **Treatment** |  | **(°C)** | **(‰)** | **(mg/L)** |
| **Control** | 7.99 ± 0.03 (n = 45) | 21.99 ± 0.93 (n = 195) | 34.31 ± 0.63 (n = 195) | 6.73 ± 0.87 (n = 195) |
| **Sodium Molybdate** | 7.99 ± 0.03 (n = 43) | 22.04 ± 0.89 (n = 195) | 34.27 ± 0.66 (n = 195) | 6.67 ± 0.81 (n = 195) |

larger in shell area than those in the control conditions (S7 Table). A t-test demonstrates this increase in shell size is significant (T-test: t = -6.2312, df = 20.65, p-value = 3.789e-06). We noticed a large difference in size variation of oysters between the two treatment buckets (Fig 2) so we applied a mixed linear model to test the effect of bucket ID on shell area for 38 day old oysters. ~55% of the variance was accounted for based on bucket ID, while ~44% of the variance remained unexplained by the model (Table 2). Thus, oysters exposed to sodium molybdate were larger in shell area than the control, and buckets played a large role in explaining the variation between individuals within treatments. As sodium molybdate was added to each treatment bucket separately, we suspect this was the primary cause of variation between treatment buckets, which contributed to observed variation in oyster growth. At 51 days molybdate-treated oysters were 31% larger in shell area (T-test: t = -2.4795, p-value = 0.02613) (Fig 2). There is a notable decrease in size difference between control and treatment animals at

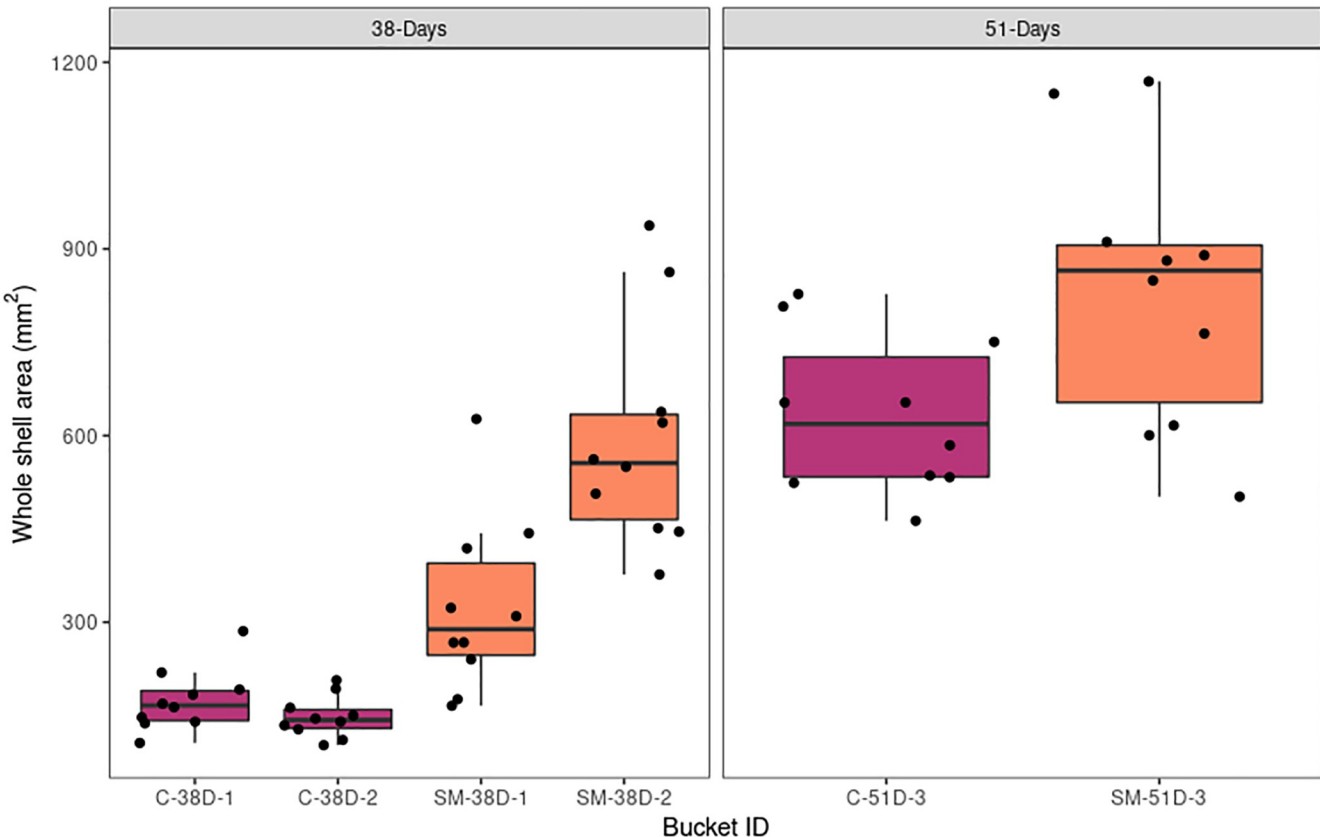

**Fig 2. Effect of sodium molybdate on oyster shell area.** Boxplots of whole shell area for control and sodium molybdate oysters collected at 38 days post-fertilization (left) and 51 days post-fertilization (right). N = 10 for each bucket.

**Table 2. Summary of linear mixed effects model fit to whole shell area (mm²) for 38 day-old oysters.**

| Fixed effects (treatment): | | | | |
|---|---|---|---|---|
| Effect | Estimate | Std. Error | t-value | p-value |
| Intercept (control) | 160.73 | 96.41 | 1.667 | 0.237 |
| molybdate | 298.69 | 136.34 | 2.191 | 0.160 |

| Random Effects: | | | | |
|---|---|---|---|---|
| Grouping | Effect | Variance | % Variance | Std. Dev |
| bucket ID | Intercept | 17004.2 | 54.63 | 130.4 |
| Residual | | 13927 | 44.75 | 118.0 |

51 days compared to 38 days. Since we did not have bucket replicates at 51 days, we could not separate the effect of treatment and bucket on size.

On a mass basis, after 38 days, oysters in the molybdate treatment group were on average ~106% heavier than those in the control (Fig 3, S8 Table). This difference in mass was statistically significant (T-test: t = -2.8201, df = 29.512, p-value = 0.008495). Mixed models found that ~12% of the variance was accounted for based on bucket ID and ~86% of the variance remained unexplained by the model, suggesting that variation between treatment buckets had less of an effect on mass than shell area (Table 3). At 51 days post-fertilization molybdate oysters were 105% heavier than control oysters, but this difference was no longer significant (T-test: t = -2.1547, p-value = 0.05716) (Fig 3).

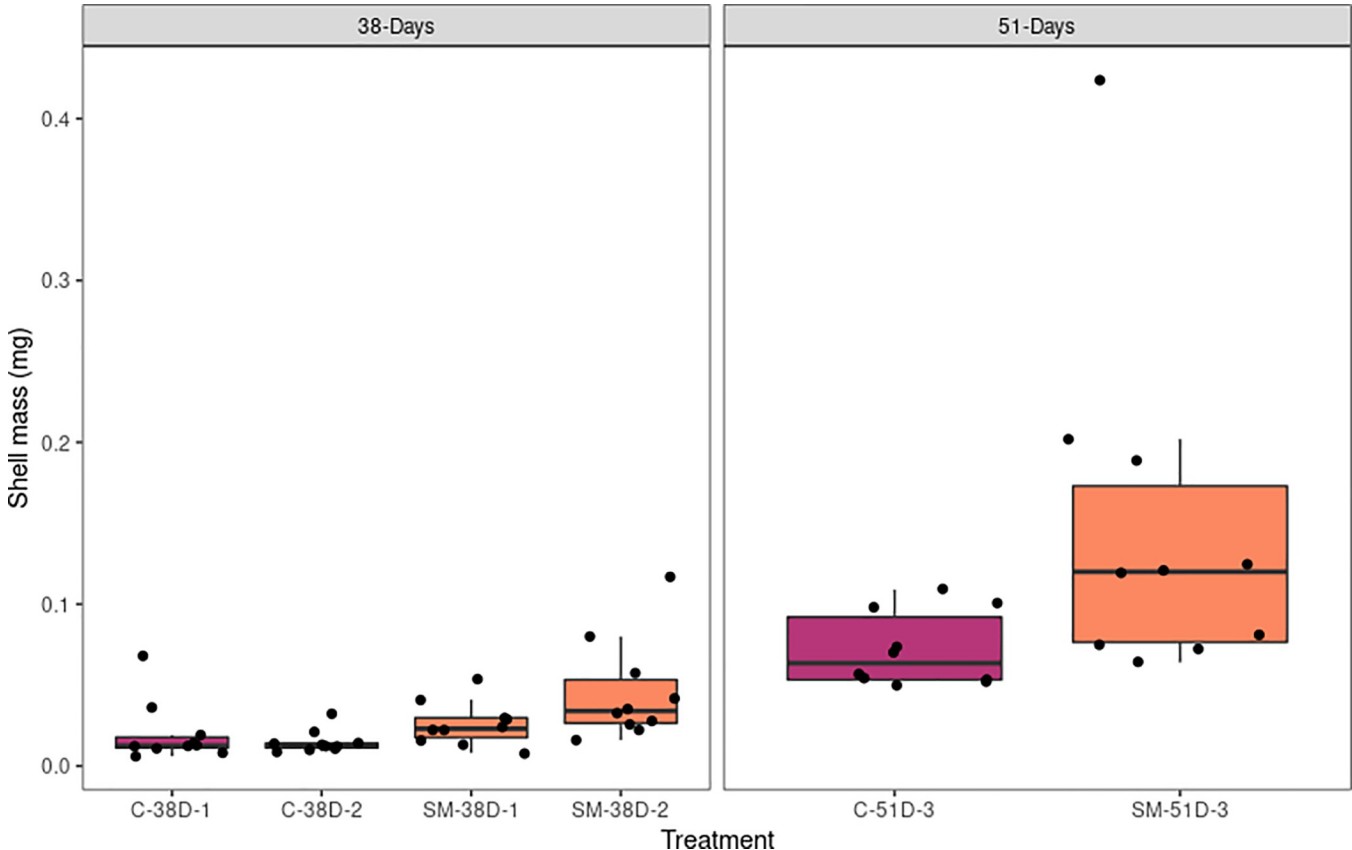

**Fig 3. Effect of sodium molybdate on oyster shell mass.** Boxplots of juvenile shell mass (mg) for control and sodium molybdate oysters collected at 38 days post-fertilization (left) and 51 days post-fertilization (right). N = 10 for each bucket.

**Table 3. Summary of fixed and random effects from linear mixed effects model fit to shell mass (mg) as the outcome variable for 38 day-old juvenile oysters only.**

Fixed effects (treatment):

| Effect | Estimate | Std. Error | t-value | p-value |
|---|---|---|---|---|
| Intercept (control) | 0.017350 | 0.007195 | 2.412 | 0.137 |
| molybdate | 0.018400 | 0.010175 | 1.808 | 0.212 |

Random Effects:

| Grouping | Effect | Variance | % Variance | Std. Dev |
|---|---|---|---|---|
| bucket ID | Intercept | 5.328e-05 | 11.68 | 0.007299 |
| Residual | | 3.918e-04 | 85.89 | 0.019795 |

The size of shells was used to estimate average growth rates through the experiment (Fig 4). Daily average growth rates in the larval shell (prodissoconch II) are similar across treatments. For the juvenile shell collected at 38 days (dissoconch) the daily average growth rate is much higher for sodium molybdate treated animals than the control. The disparity between treatments appears to decrease for animals collected at 51 days. Taken collectively, our data suggests that size differences between treatments was driven by an early phase of increased growth in sodium molybdate-treated oysters, and that the rate of growth converged as the animals approached 51 days.

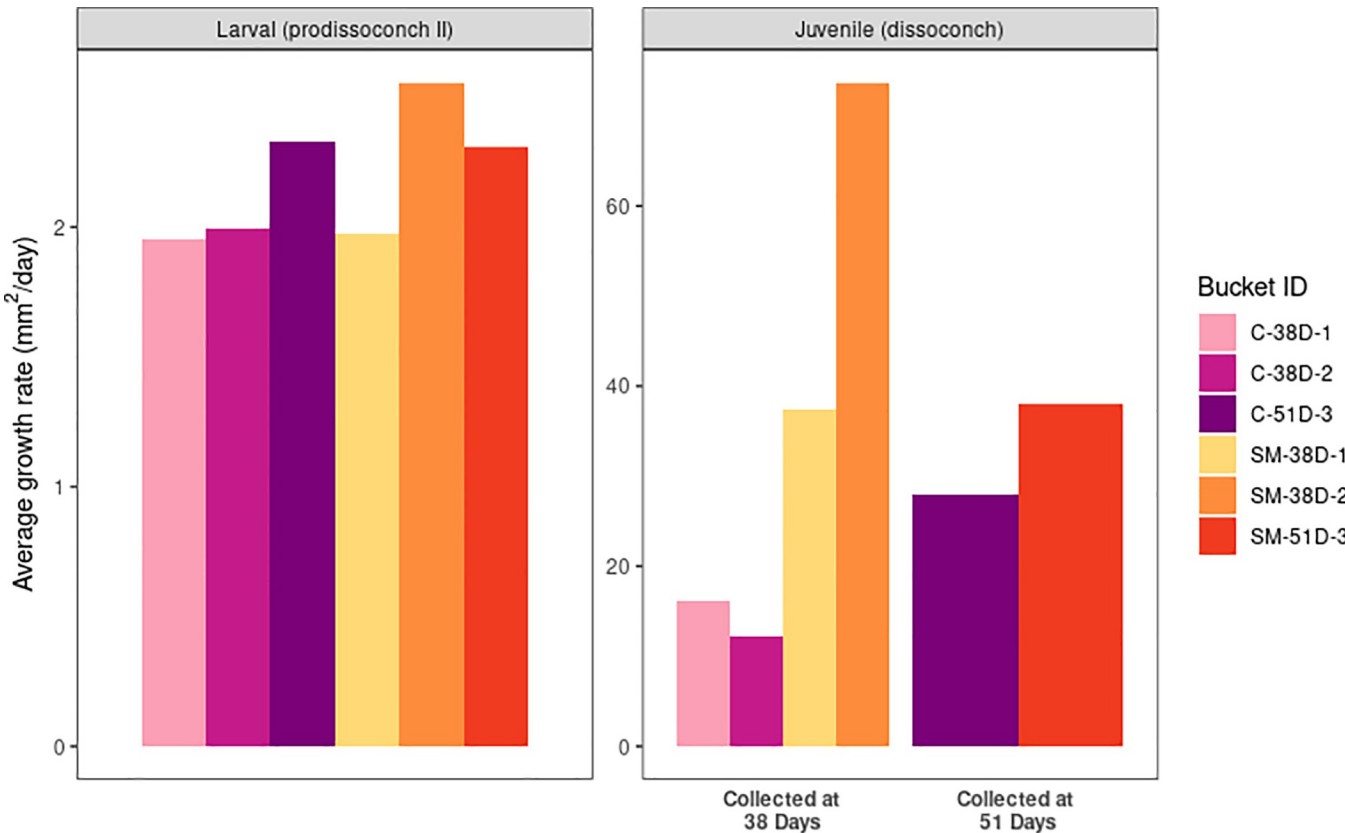

**Fig 4. Estimated daily average growth rate (mm²/day) for each bucket analyzed.** Mean shell area was calculated for the larval shell (prodissoconch II) and juvenile shell (dissoconch) for each bucket. Larval shell area was divided by 31 days (approximate date of settlement) for all buckets to estimate daily mean larval shell growth rate. To estimate daily mean juvenile shell growth rate, mean juvenile shell sizes were divided by seven day and 20 days if they were harvested on day 38 or day 51 post-fertilization, respectively. Calculated growth rates can be found in S9 Table.

## Patterns and differences in microbial community composition

Alpha diversity did not differ among buckets from the molybdate and control treatments (Kruskal-Wallis; Shannon: $X^2 = 10.978$, df = 5, p = 0.05182; Observed: $X^2 = 8.3897$, df = 5, p = 0.136; Fig 5), but beta diversity (i.e. inter-sample diversity) varied significantly between treatments (PERMANOVA, $F_{1,14} = 13.318$, $p_B = 0.016$) (Table 4, Fig 6). When samples were subgrouped by treatment and date of collection (38 versus 51 days), there were also significant differences between groups (PERMANOVA, $F_{3,12} = 6.8354$, $p_B = 0.002$) (Table 4, Fig 6). Using a more stringent Bonferroni p-value correction, only the comparison between treatments at 38 days remained statistically significant (PERMANOVA, $F_{1,8} = 6.2971$, $p_B = 0.032$) (Table 5, Fig 6). No comparisons yielded a significant p-value for Levene's test for homogeneity of dispersions, indicating that groups had similar within-group variance and that PERMANOVA

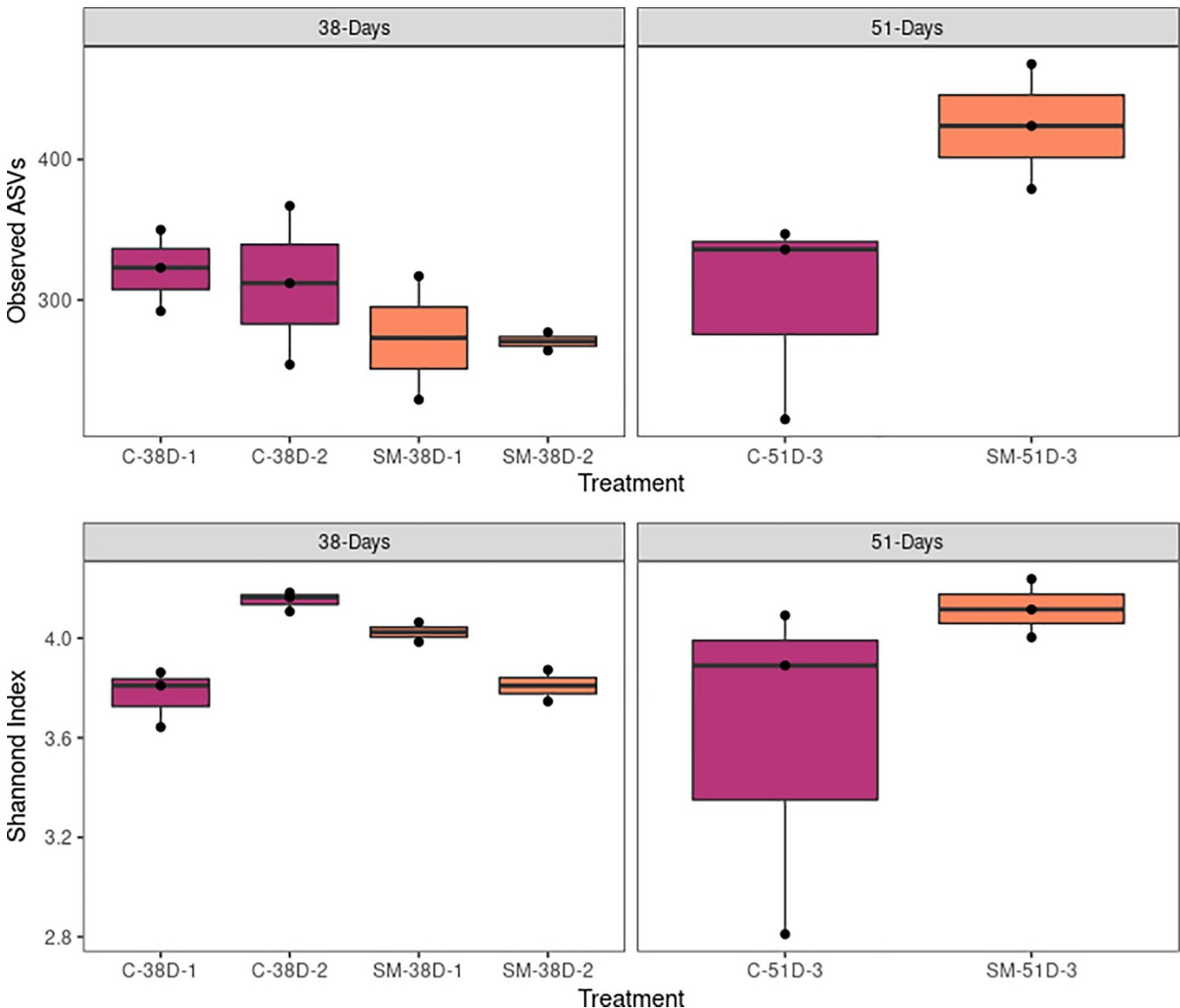

**Fig 5. Observed ASVs and Shannon index were calculated for each sample.** C: Control; SM: Sodium Molybdate. N = 3 for each bucket, except SM-1 and SM-2, for which N = 2.

**Table 4. Summary of PERMANOVA and Levene's test for homogeneity of dispersions.**

| Comparison | PERMANOVA | | | | Levene's test | | |
|---|---|---|---|---|---|---|---|
| | Pseudo-F | $R^2$ | p-value | $p_B$ | F | p-value | $p_B$ |
| Treatments | 3.1197 | 0.18223 | 0.008 | 0.016* | 0.002 | 0.974 | 1.000 |
| Treatments-Time | 6.8354 | 0.63084 | 0.001 | 0.002* | 3.2811 | 0.37 | 0.74 |

This table includes tests comparing Control and Sodium Molybdate treatments, and all time points within each treatment. A significant p-value for PERMANOVA indicates that the groups tested are statistically distinct. A significant p-value for Levene's test indicates that the group dispersions of the groups tests are statistically different, which breaks one of the assumptions of PERMANOVA. Bonferroni adjusted p-values ($p_B$) have been corrected for k = 2 tests, and (*) indicates $p < 0.05$.

results were valid (Table 5). Overall, these results suggest that the oyster microbiome was significantly different between treatments at 38 days post-fertilization, but by 51 days the oyster microbiomes were no longer statistically distinct.

The addition of molybdate to treatment buckets had a significant, measurable effect on the relative abundance of bacterial orders present in samples. After 38 days post-fertilization, the abundances of Thiohalorhabdales, Clostridiales, and Thermoanaerobaculales were significantly higher in the control than the sodium molybdate treatment, whereas Caulobacterales and Arenicellales were significantly more abundant in the molybdate treatment (Fig 7, S6 Table). By 51 days post-fertilization Babeliales was the only order that had significantly different abundances between treatments, and was higher in molybdate treated oysters (Fig 7, S6 Table).

### Presence of sulfate-reducing bacteria

Most sulfate-reducing bacteria are members of clade Deltaproteobacteria, a group that represented less than 1.5% of the bacteria in all of our samples. Of the Deltaproteobacteria Families found in our samples, only Desulfarculaceae and Desulfobulbaceae are known sulfate-reducers (Fig 7). Desulfobulbaceae was found in low abundance in sample C-51D-3, while Desulfarculaceae was found in greater abundance in SM-38D-1, SM-38D-2 and SM-51D-3 (Fig 8). It was notable that Desulfarculaceae was present in SM-38D-2 and SM-38D-3, but not SM-38D-1 (Fig 8). However, this is not necessarily surprising given that Desulfarculaceae are rare and both SM-38D-1 and SM-38D-2 had fewer 16S reads than SM-38D-3 (S2 Table). Overall, this indicates that the sodium molybdate did not inhibit sulfate-reducing Deltaproteobacteria and in fact may have promoted Desulfarculaceae in oysters exposed to sodium molybdate.

In addition to Deltaproteobacteria, some sulfate-reducers have been described in the Clostridiales and Thermoanaerobaculaceae [55,56]; we therefore further analyzed these orders to see whether putative sulfate reducers demonstrated differential abundance between treatments. All Thermoanaerobaculales in our dataset were members of the family Thermoanaerobaculaceae, which are not known sulfate reducers. Christensenellaceae did include members of the Christensenella, Lachnospiraceae, and Ruminococcaceae, which have putative sulfate reducers (Fig 8) [57,58]. At 38 days post-fertilization, most of these clades were absent from sodium molybdate treated animals, although Ruminococcaceae were present in one molybdate bucket (Fig 8). At 51 days post-fertilization, the difference between treatments was no longer statistically significant. Taken with the data from Deltaproteobacteria, our results suggest that sodium molybdate might have a negative effect on some sulfate reducers, particularly those from the family Christensenella, but those effects appear to be temporary.

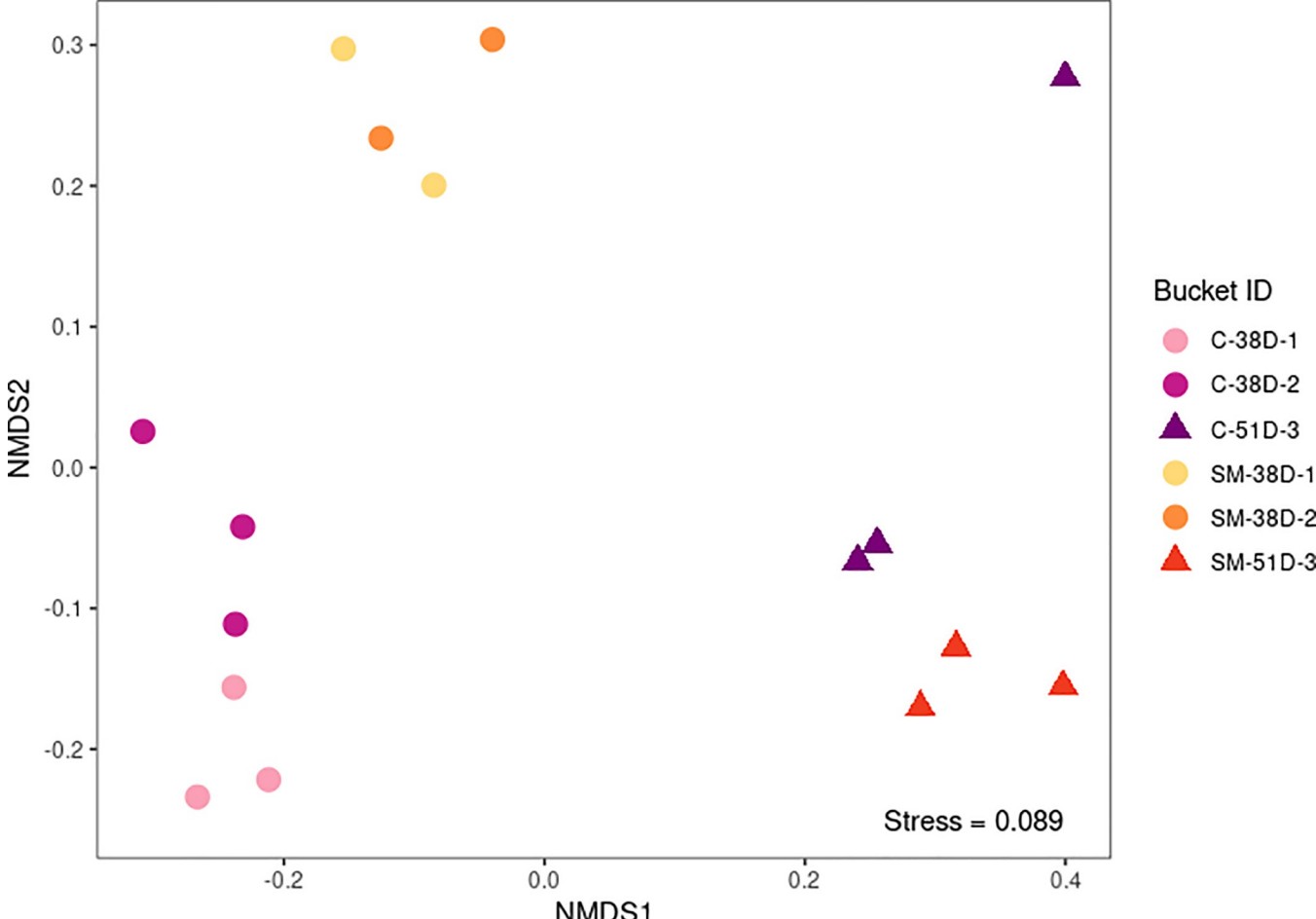

**Fig 6. Non-metric multidimensional scaling visualization based on Bray-Curtis dissimilarities of bacterial communities.** Resultant stress of the NMDS ordination was 0.089, indicating that the 2-D NMDS plot is a good representation of Bray-Curtis distances between buckets [54].

## Different gene expression analysis

Similar to the microbiome, host gene expression analysis suggests that treatments were more different from each other at 38 days post-fertilization than 51 days. Principal component analysis (Fig 9) suggests that at 38 days post-fertilization, oysters exposed to sodium molybdate showed distinct expression signatures from those in the control treatment. At 51 days, biological replicates

**Table 5. Summary of pairwise post hoc PERMANOVA and Levene's test for homogeneity of dispersions.**

| | PERMANOVA | | | | Levene's test | | |
|---|---|---|---|---|---|---|---|
| Comparison | Pseudo-F | $R^2$ | p | $p_B$ | F | p | $p_B$ |
| C-38D-[1+2]:C-51D-3 | 6.4619 | 0.48002 | 0.018 | 0.072 | 0.492 | 0.497 | 1.000 |
| C-38D-[1+2]:SM-38D-[1+2] | 6.2971 | 0.44045 | 0.008 | 0.032* | 4.0024 | 0.063 | 0.252 |
| C-51D-3:SM-51D-3 | 3.608 | 0.47423 | 0.1 | 0.4 | 0.338 | 0.7014 | 1.000 |
| SM-38D-[1+2]:SM-51D-3 | 8.8303 | 0.63847 | 0.028 | 0.112 | 3.3928 | 0.138 | 0.552 |

This table includes tests between all time points within each treatment.

(*) means $p < 0.05$. Bonferroni adjusted p-value ($p_B$), k = 4.

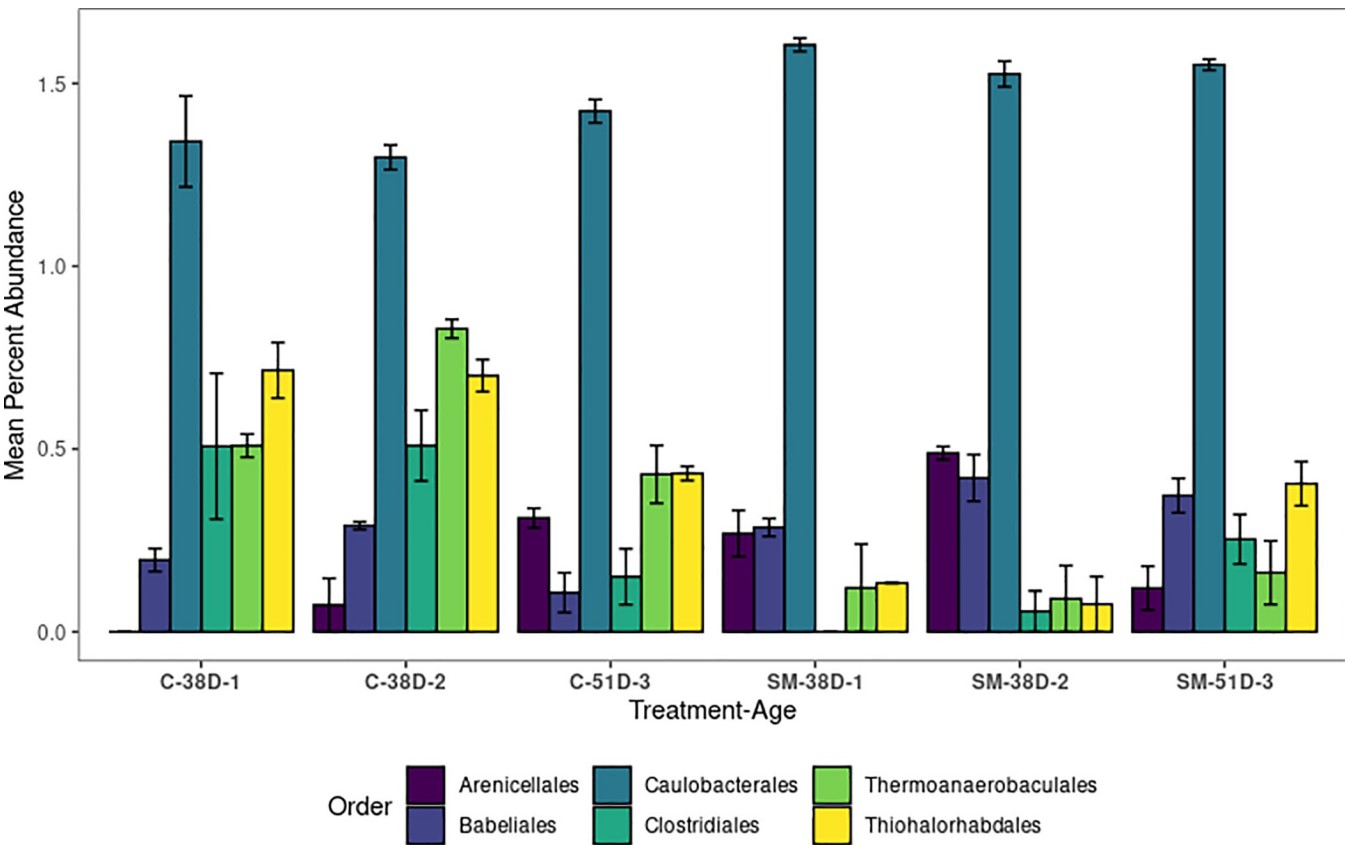

**Fig 7. Differences in microbial community structure across treatments and time.** The mean percent abundance of ASVs, colored by taxonomic Orders, that had significantly different abundances (Dunn test, $p_b < 0.05$) between the control and molybdate treatments at either 38 or 51 days post-fertilization. Bars are colored by Order identity, and error bars represent one standard error.

across the two treatments no longer formed distinct clusters. This suggests that oyster gene expression profiles were strikingly similar 51 days post-fertilization between treatments.

To interpret these differences, we performed differential gene expression analysis with cuff-diff, and tested for enriched gene ontology (GO) terms. The results of these analyses are summarized in S10 Table. Most comparisons between samples produced few if any enriched GO terms. At 38 days post-fertilization, most significant differences between the control and sodium molybdate treatments came from peptidase-related activity (Fig 10). At 51 days there are no enriched GO terms between the two treatments (Fig 10). The largest number of enriched GO terms comes from comparing the sodium molybdate treatment at 38 versus 51 days, which is dominated by changes in metabolism as well as biological responses to other organisms. This latter set of GO terms is particularly intriguing, as it suggests the organism could be changing its gene expression profile to respond with changes in the microbial community. The hierarchical clustering in Fig 10 suggests the genes can be broadly divided into those that are upregulated at 51 days and those that are downregulated. Detailed annotation of the genes in the two clusters is provided in S11 Table.

## Discussion

The primary goal of this experiment was to characterize the effects of sodium molybdate on the Pacific oyster *M.gigas*. Our *a priori* hypothesis was that sodium molybdate would

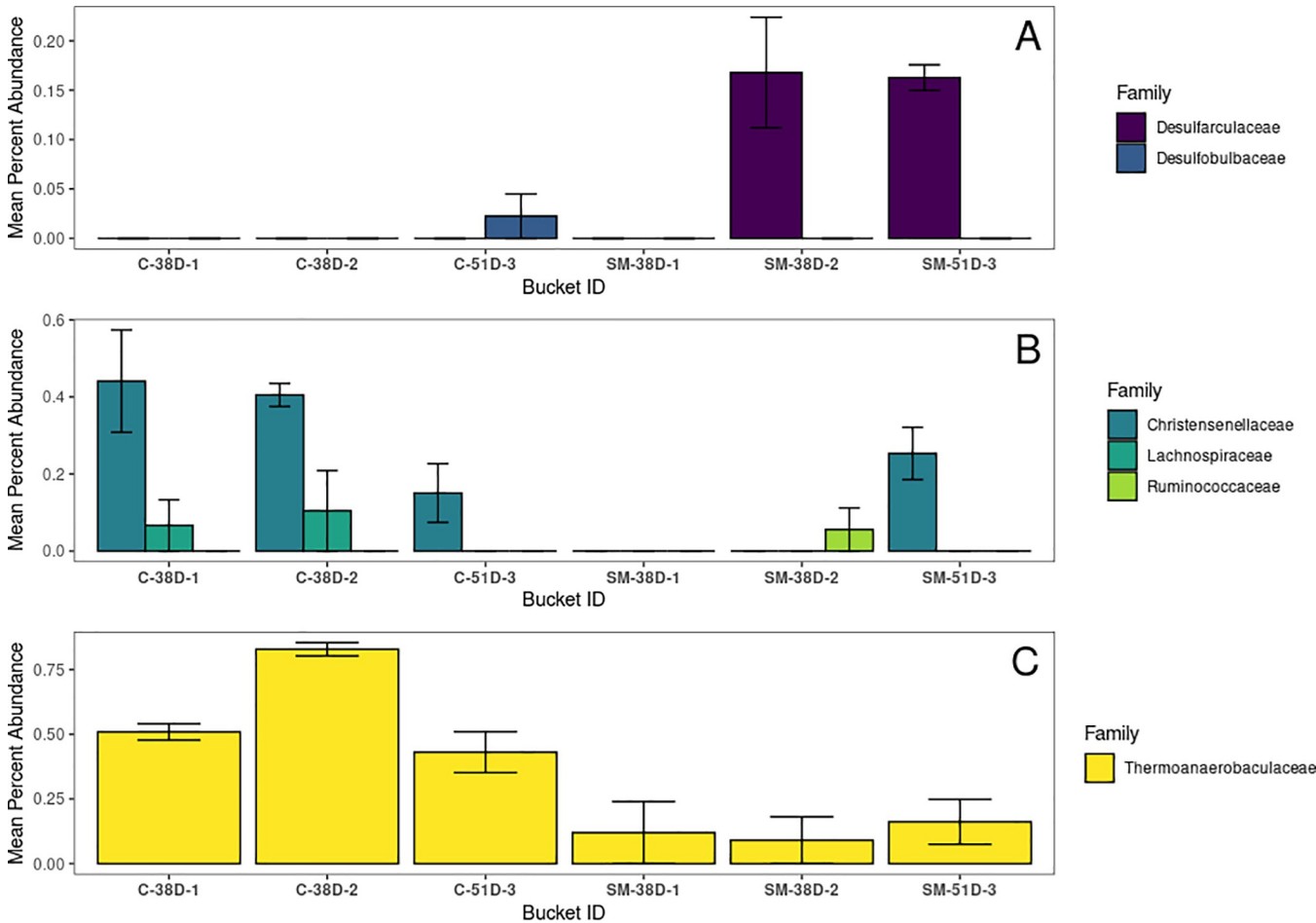

**Fig 8. The mean percent abundance of ASVs colored by taxonomic family.** A) from the Class Deltaproteobacteria, B) the Order Clostridiales, and C) the Order Thermoanaerobaculales, all of which include known sulfate-reducers. Each bucket within treatments is represented by 3 samples taken from that bucket (i.e. n = 3 for each bucket), except SM-38D-1 and SM-38D-2, for which n = 2. Error bars represent one standard error.

competitively replace sulfate ions, inhibiting activity by sulfate-reducing bacteria and potentially leading to decreased oyster growth due to the influence of these bacteria on calcification [27,28]. However, oysters grew larger when exposed to sodium molybdate compared to the control group (Figs 2 and 3), contradicting our initial hypothesis. In a previous paper using sodium molybdate on adult oysters, [25] found that sodium molybdate treated animals had shells that were significantly more dense than control animals (with mean densities of 2.31 gcm$^{-3}$ and 1.92 gcm$^{-3}$, respectively). Although the shells in the present study were too small to perform the same test on density, our results are consistent with growing evidence that sodium molybdate enhances shell density; this would explain why the sodium-molybdate treated shells remained comparably heavier than controls even as differences in shell area decreased over time.

Regarding the mechanism sodium molybdate plays in increasing shell growth, our results suggest it is not due to the elimination of sulfate reducing bacteria. Relative abundance of sulfate-reducing Deltaproteobacteria increased after exposure to sodium molybdate. Some rarer sulfate reducers, particularly members of the Christensenella, may have been downregulated by sodium molybdate, but if so the effect appears temporary. Previous studies evaluating the inhibitory effect of sodium molybdate on microbial sulfate reduction in pure cultures has

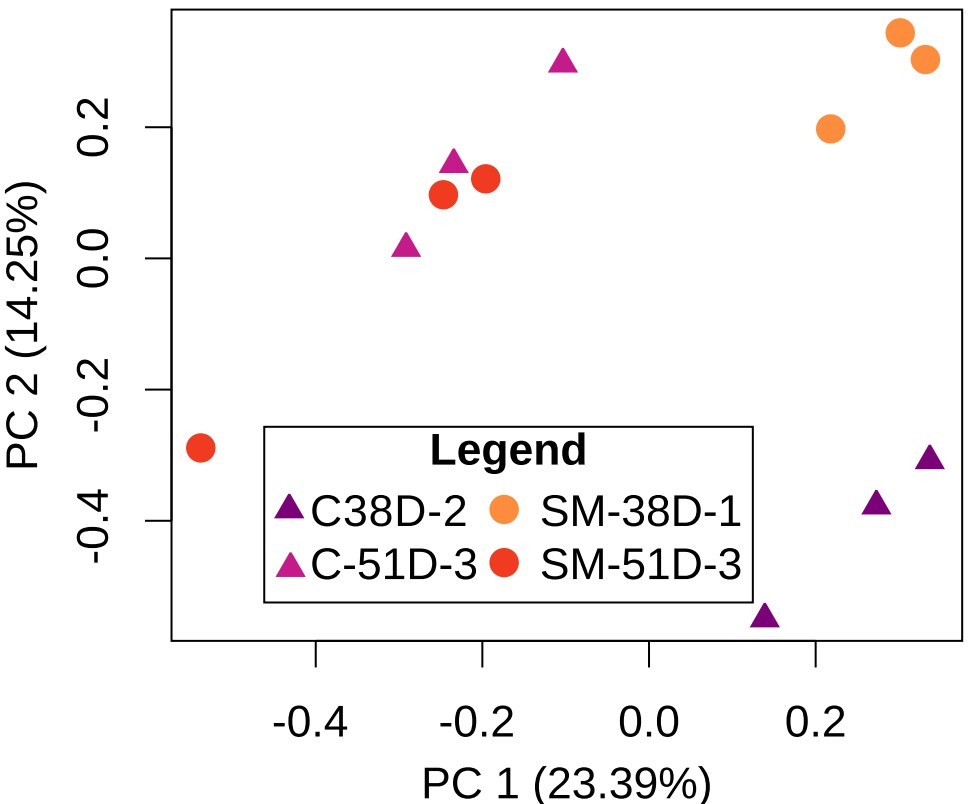

**Fig 9. PCA plot visualizing similarity of gene expression profiles between replicates.** The plot was calculated based on fpkm (fragment per kilobase per million reads) adjusted read counts, log transformed and centered around the mean.

shown that different taxa show varying levels of inhibition at a given molybdate concentration [59]. Whether or not bacterial sulfate reduction produces conditions that enhance carbonate precipitation is dependent on the terminal electron donor used during metabolism, which are also taxon specific [60,61]. Thus, changes in bacterial taxonomic composition, particularly of putative sulfate-reducers in Thermoanaerobaculales, Clostridiales, Desulfoarculaceae, may have affected chemistry in the calcifying fluid of the oysters or other elements therein that play a role in shell formation. Finally, while this paper focused on the role of sulfate-reducing bacteria, we recognize that microbes of other metabolic types (e.g. denitrification, ammonification) can also influence host health and development. Further work is required to address how sodium molybdate impacts the microbial community outside of sulfate reducers, and whether different microbial metabolisms affect shell formation. Overall our results suggest that the presence of sodium molybdate does not generically inhibit sulfate-reducing bacteria in oysters, but does have taxon-specific effects on the microbiome and enhances shell calcification.

Many of the differences observed between sodium molybdate and control oysters at 38 days post-fertilization diminished by 51 days. While shell area was significantly higher in sodium molybdate-exposed oysters at both time points, the magnitude of this difference was smaller at 51 days (Fig 2). Bacterial community composition (beta diversity) varied between treatments at 38 days post-fertilization but no longer differed by 51 days (Table 5, Fig 6). Finally, transcriptome analyses suggest that gene expression profiles of oysters exposed to sodium molybdate are distinct from the control at 38 days, but overlap by 51 days (Fig 9). Most enriched GO terms were found when comparing sodium molybdate treated time points, and were primarily

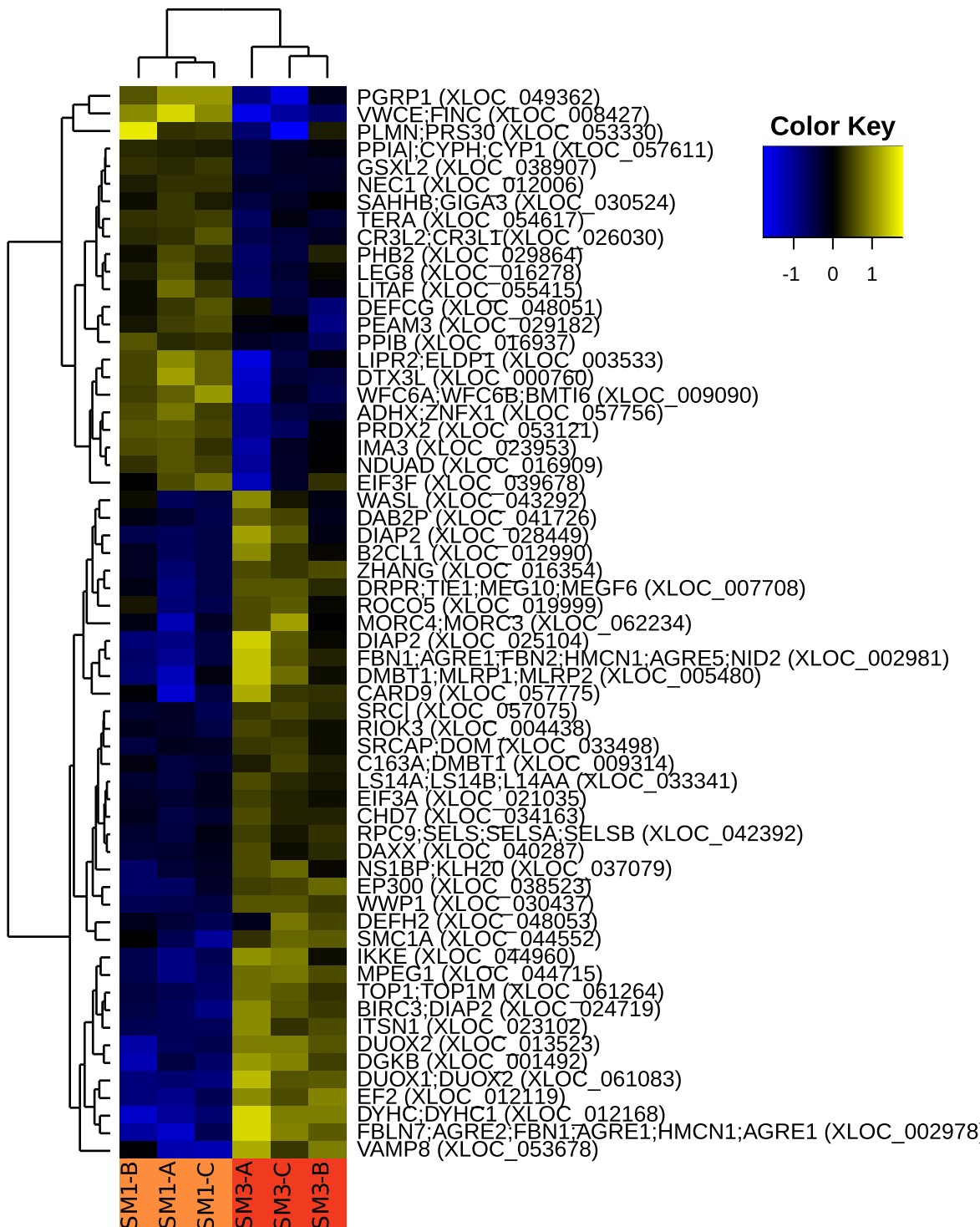

**Fig 10. Heat map demonstrating the expression of *M. gigas* genes related to host response to other organisms.** The dendrogram on the left side of the heat map demonstrates the similarity of genes to each other based on their relative expression; the dendrogram at the top demonstrates the similarity between samples. Names to the right of each row start with a gene ID based on Uniprot identifiers; more than one ID is included if different isoforms from the same gene matched to different Uiprot genes. The names on the right also include the ID of the relevant *M. gigas* transcript from our transcriptome assembly. GO terms used for this analysis include: "Response to other organism" (GO:0051707), "response to external biotic stimulus" (GO:0043207), "defense response to other organism" (GO:0098542), "multi-organism process" (GO:0051704), and "response to biotic stimulus" (GO:0009607). The values for each gene were calculated based on fpkm adjusted read counts, log transformed and centered around the mean.

related to metabolism and responses to other organisms (i.e. bacteria) (S10 Table). Focusing in on the genes related to other organisms (Fig 10, S11 Table), there is an intriguing nexus between antimicrobial activity, shell formation, and host immunity. Some notable candidate proteins from our dataset include VWCE and GIGA3, which are known shell matrix proteins, as well as CARD9, DIAP2, and PGRP, which are associated with innate immunity [62–64].

This is consistent with previous work illustrating that while the innate immune system of marine invertebrates applies top-down selection on microbial partners, the microbiome also affects host health, functioning, and physiology [65–67]. Moreover it is well documented that aspects of the oyster innate immune system (e.g. hemocytes) are responsible for directing aspects of shell formation [68–72]. Our interpretation of these observations is that oysters exposed to sodium molybdate were adjusting their gene expression profiles to respond to dysbiosis in the microbial community, creating a compensatory mechanism that limited differences between animals by 51 days. This hypothesis might seem counterintuitive as oysters exposed to sodium molybdate grew larger shells, but recent research suggests that environmental stressors can cause young oysters to invest in shell growth to the detriment of biomass [73]. While genetic regulation by the host against dysbiosis potentially explains the various results of this study, it remains unclear whether sodium molybdate changes to the microbiome or shell growth were actually disadvantageous to oyster fitness. Regardless, the evidence presented here indicates that complex interactions between oyster gene expression and the microbiome can have a significant effect on organismal processes such as shell formation.

## Conclusions

In the present study, we reared Pacific oysters to 51-days post fertilization and exposed them to control and sodium molybdate conditions. Molybdate-exposed oysters precipitated significantly more shell material (as measured by shell area and mass) than control oysters. Exposure to sodium molybdate did not have the predicted effect, which was that molybdate would display a whole-sale inhibitory effect on sulfate-reducing bacteria regardless of taxonomic identity and decrease shell formation. In contrast, results showed that the abundance of *Desulfarculaceae* actually increased in oysters exposed to molybdate. This indicates that there are more complex mechanisms that control microbial community structure and composition than was initially anticipated. Additionally, results show that there were greater differences in oyster microbiome and transcriptome between treatments early in ontogeny (38-days post settlement), but became more similar at later developmental stages (51-days post settlement). We hypothesize that these changes can be understood through host-based compensation, changing gene expression to adjust the microbiome and ultimately shell growth.

## Supporting information

**S1 Table. Summary of project activity and milestones over the course of the experiment.** "Spec pH" refers to water samples collected to be analyzed on the Ocean Optics Jaz Spectrophotometer.
(XLSX)

**S2 Table. pH values measured on the Ocean Optics Jaz Spectrophotometer (pH.uncorrected), with the tris buffer offset correction (pH.trist.correct).** Alkalinity values were measured on Gran titration on a Metrohm 809 Titrando. Mean alkalinity was calculated for each bucket on the basis of samples that were successfully analyzed for that bucket and mean total alkalinity values for each bucket were assigned to each spectrophotomically derived pH value that also belonged to the same bucket (avg.alk). pH was finally corrected using these alkalinity

values in the CO2Sys calculator (Robbins et al., 2010). InCntrl refers to measurements taken on seawater used to restock control and molybdate tanks after each water change.
(XLSX)

**S3 Table. Summary of sequence counts after each bioinformatics step in DADA2 and phyloseq.**
(XLSX)

**S4 Table. Mean, standard deviation, and standard error of the mean percent abundances of ASVs as taxonomic Orders across all treatments.** Grouped on the basis of treatment-time.
(XLSX)

**S5 Table. Results of comparisons made using the Kruskal-Wallis test between treatment-time groups (C-38D-1 + C-38D-2, C3-51D-3, SM-38D-1 + SM-38D-2, SM-51D-3) for ASVs as taxonomic Orders.**
(XLSX)

**S6 Table. Pairwise post-hoc Dunn tests comparing mean percent abundance of ASVs as taxonomic Orders between age cohorts within the Molybdate and control treatment groups.**
(XLSX)

**S7 Table. Shell area data in micrometers^2 measured in Image J from SEM images of shells.** Whole refers to the entire shell area (prodissoconch II + dissoconch).
(XLSX)

**S8 Table. Shell mass data (mg) for individual oysters from each bucket.**
(XLSX)

**S9 Table. Mean growth rate for oysters by bucket.** Larval growth rate represents prodissonconch II growth (0–31 days) and juvenile growth rate represents dissoconch growth (31–38 and 51 days).
(XLSX)

**S10 Table. Results from differential gene expression analysis with cuffdiff and enriched gene ontology (GO) terms.**
(XLSX)

**S11 Table. Overview of differentially expressed genes driving enrichment of GO terms involved in responses to other organisms.** Annotations are based on best BLAST hits against the UNIPROT/SWISSPROT database. Note that some genes had multiple isoforms with different best hits; in those cases multiple annotations are provided.
(XLSX)

## Acknowledgments

Thank you very much to Hog Island Oyster Company for supplying us with animals for this project. Thanks also to Karl Menard and the Aquatic Resources Group at the Bodega Marine Laboratory, and especially Joe Newman who lent his time and expertise to the project. Thank you to Dr. Tessa M. Hill who allowed us to use instruments and space in her lab, and to Sarah Merolla who provided technical assistance running the machines. Thank you to Dr. David Coil, who provided feedback on an early version of this manuscript. Thanks also to Dominic Dickerson, who assisted with animal husbandry. And finally, thank you to our funding source, the University of California Davis Microbiome Special Research Program.

## Author Contributions

**Conceptualization:** Roxanne M. W. Banker, John J. Stachowicz, David A. Gold.

**Data curation:** Roxanne M. W. Banker, David A. Gold.

**Formal analysis:** Roxanne M. W. Banker, Jacob Lipovac, David A. Gold.

**Funding acquisition:** Roxanne M. W. Banker, John J. Stachowicz, David A. Gold.

**Investigation:** Roxanne M. W. Banker.

**Methodology:** Roxanne M. W. Banker, David A. Gold.

**Supervision:** David A. Gold.

**Visualization:** Roxanne M. W. Banker.

**Writing – original draft:** Roxanne M. W. Banker, David A. Gold.

**Writing – review & editing:** Roxanne M. W. Banker, John J. Stachowicz, David A. Gold.

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
