## [Decision Letter · Decision Letter 0]

5 Nov 2021

PONE-D-21-29631Sodium molybdate does not inhibit sulfate-reducing bacteria but increases shell growth in the Pacific oyster *Magallana gigas*PLOS ONE

Dear Dr. Banker,

Thank you for submitting your manuscript to PLOS ONE. After careful consideration, we feel that it has merit but does not fully meet PLOS ONE’s publication criteria as it currently stands. Therefore, we invite you to submit a revised version of the manuscript that addresses the points raised during the review process.

We look forward to receiving your revised manuscript.

Kind regards,

José A. Fernández Robledo, Ph.D.

Academic Editor

PLOS ONE

Journal Requirements:

2. Please note that in order to use the direct billing option the corresponding author must be affiliated with the chosen institute. Please either amend your manuscript to change the affiliation or corresponding author, or email us at plosone@plos.org with a request to remove this option.

Additional Editor Comments:

Dear Dr. Banker,

let me first apologize for the time that took to send you back the reviews, I have had difficult finding reviewers willing to review your manuscript. I have reviewed it my self as a non specialist on the subject and I have no major comments and some of the edits are already indicated for the other reviewer. Overall, I find the experimental procedures and presentation of results were very clear. The oysters spat was produced following hatcheries procedures. Still, the reviewer rises some questions for you to address regarding, e.g., "the experimental setup was not oxygen-deprived, and often sulfate-reducing prokaryotes thrive in anaerobic environments." Please, address the concerns.

Line 188, add the reference for the primers used.

Take also the time to review the references as some of them are not in the same format (e.g., capitalization of the title, journal abbreviation, italics in scientific names).

-j

Reviewers' comments:

Reviewer's Responses to Questions

**Comments to the Author**

1. Is the manuscript technically sound, and do the data support the conclusions?

Reviewer #1: Partly

2. Has the statistical analysis been performed appropriately and rigorously? 

Reviewer #1: Yes

3. Have the authors made all data underlying the findings in their manuscript fully available?

Reviewer #1: Yes

4. Is the manuscript presented in an intelligible fashion and written in standard English?

Reviewer #1: Yes

5. Review Comments to the Author

Reviewer #1: Overall Comments:

The main goal of this manuscript was to characterize the effects of sodium molybdate on the Pacific oyster M.gigas based on the fact that sodium molybdate would competitively replace sulfate ions, inhibiting activity by sulfate-reducing bacteria (SRBs) and potentially leading to decreased oyster growth due to the influence of these bacteria on calcification. But this manuscript lacks a critical component when it comes to sulfate-reducing bacteria and the experimental setup. The experimental setup was not oxygen-deprived, and often sulfate-reducing prokaryotes thrive in anaerobic environments. This is evident from the presence of SRBs in low abundances in the system. Since SRBs are present in such low abundances in the system, it is hard to reach any conclusion. Microbiome harvesting, DNA extractions, and sequencing depth will have a huge impact on these lower abundance data. The changes in different taxa can also be contributed by the presence of multiple copies of the 16S rRNA gene in a bacterium which was not accounted for in this study.

Other critical questions are

1) Do all SRBs promote calcification?

2) Is there any other microbial population in the system which can affect oyster growth or promote calcification?

Other microbial populations which were abundant in the system were not reported/discussed in the manuscript. Those microbial populations can also play role in promoting the growth of Oysters. So, the antagonistic behavior of sodium molybdate could have been countered by the activity of other microbial populations. I feel that the starting microbial community will also be a determining factor for supporting/contradicting the hypothesis.

Specific comments:

Line 49: “A previous study showed ……” in place of “[21] showed”. Then the reference should be added at the end of the sentence.

Line 61: 1500 L

Line 78: 20 µm

Line 82: What were the criteria for selection based on egg quality?

Line 145: 35 µm

Line 181: Elaborate the method heading

Line 181: S will be in uppercase for 16S; here and in other places.

Line 205: Add reference for Decontam’s prevalence method.

Line 210: >25,000 ASVs or a total of >25,000 ASV abundances?

Line 223: Bray-Curtis dissimilarity

Line 532: What does concentration of bacteria mean? Will it be abundance?

6. PLOS authors have the option to publish the peer review history of their article (what does this mean?). If published, this will include your full peer review and any attached files.

Reviewer #1: No

---

## [Author Response · Author response to Decision Letter 0]

4 Jan 2022

Response to Reviewers

Editor Comments:

Line 188, add the reference for the primers used. Change made, Parada reference added.

Take also the time to review the references as some of them are not in the same format (e.g., capitalization of the title, journal abbreviation, italics in scientific names).

The references have been reviewed and fixed.

Reviewer #1: Overall Comments:

The main goal of this manuscript was to characterize the effects of sodium molybdate on the Pacific oyster M.gigas based on the fact that sodium molybdate would competitively replace sulfate ions, inhibiting activity by sulfate-reducing bacteria (SRBs) and potentially leading to decreased oyster growth due to the influence of these bacteria on calcification. But this manuscript lacks a critical component when it comes to sulfate-reducing bacteria and the experimental setup. The experimental setup was not oxygen-deprived, and often sulfate-reducing prokaryotes thrive in anaerobic environments. This is evident from the presence of SRBs in low abundances in the system. Since SRBs are present in such low abundances in the system, it is hard to reach any conclusion.

The reviewer brings up an important caveat regarding the role of SRBs in animal biomineralization. However, there is ample evidence that SRBs are capable of precipitating calcium carbonate in aerobic environments, including oxygenated sediments and microbial mats (e.g. Visscher et al. 2000; Jørgensen BB 1994; Canfield D & Des Marais DJ 1991; Cypionka et al. 2000). There are also plenty of opportunities for anaerobic microenvironments to develop in the oyster shell, either in fluid-filled spaces within the shell or during times of oyster valve closure (Vermeij 2013; 2020). Our goal was to study the role of SRBs in shell formation under natural conditions, so artificially creating anaerobic environments would not make sense in this context. We have added a new paragraph in the introduction to bring up these relevant points and references.

Microbiome harvesting, DNA extractions, and sequencing depth will have a huge impact on these lower abundance data. The changes in different taxa can also be contributed by the presence of multiple copies of the 16S rRNA gene in a bacterium which was not accounted for in this study.

We acknowledge that 16S rRNA gene copy number variability could cause certain microbial taxa to appear more abundant than is actually the case. This is a challenge for all microbiome studies that use the amplicon sequencing approach. How to correct for gene copy number remains an open question and tools developed for this purpose have been unreliable thus far (e.g. Louca et al., 2018; Stark et al., 2021). For these reasons, we chose not to correct for 16S rRNA copy number in this study. We added text and references to the Methods subsection “Microbiome Analysis and Visualization” to note this concern and justify our decision to not perform corrections.

Other critical questions are

1) Do all SRBs promote calcification?

There is compelling evidence that SRBs as a group can promote calcification, but there is uncertainty in the extent to which particular clades do so, the environments in which they can do it, and their role in animal calcification. Our new paragraph in the introduction includes these points.

2) Is there any other microbial population in the system which can affect oyster growth or promote calcification?

There are almost certainly other microbial taxa that affect oyster calcification. The entire microbial community dictates the alkalinity of the microenvironment and the subsequent probability that minerals will precipitate. While our focus in this project was on SRBs, we have added this caveat to the discussion.

Other microbial populations which were abundant in the system were not reported/discussed in the manuscript.

Those microbial populations can also play a role in promoting the growth of Oysters. So, the antagonistic behavior of sodium molybdate could have been countered by the activity of other microbial populations.

We do provide a summary report of the microbial clades that exhibit significant differences between treatments (see section “Patterns and differences in microbial community composition”, Figure 7, and Table S6). As mentioned in the previous response, it is certainly possible that some of these non-SRB clades play a role in calcification. While we have no direct evidence of that in this study, we allude to the possibility in the discussion (see our response to the previous point). We are also preparing a follow-up study that will allow us to compare the microbial community in our experimental oysters to animals in other environmental conditions that produce similar changes in shell structure. That has the potential to reveal other candidate microbes that are broadly associated with shell growth. We appreciate the Reviewer’s suggestion for this analysis.

I feel that the starting microbial community will also be a determining factor for supporting/contradicting the hypothesis.

We do have microbiome data from pre-settlement larvae. However, the microbial community in larvae is dramatically different from post-settlement oysters, and provides little information on “starting” conditions relevant to the interpretation of the experiment. Transcriptome and 16S data from the larval treatment are publicly available through our NCBI Bioproject, and we anticipate discussing this data in a follow-up manuscript. Despite these limitations, we feel the control population provides sufficient insight for us to test the impact of sodium molybdate on the oyster microbial community. If the reviewer wishes to expand on how they think our hypotheses could be strengthened with information from the “starting” community, we would be happy to consider it.

Specific comments:

Line 49: “A previous study showed ......” in place of “[21] showed”. Then the reference should be added at the end of the sentence.

Change made.

Line 61: 1500 L

Change made.

Line 78: 20 μm

Change made.

Line 82: What were the criteria for selection based on egg quality?

Text added: Eggs were considered good quality if they had a regular, round shape with a plump appearance, as opposed to ovate or wrinkly.

Line 145: 35 μm

Change made.

Line 181: Elaborate the method heading

Changed to “16S rRNA Sample Preparation and Sequencing”

Line 181: S will be in uppercase for 16S; here and in other places.

Change made.

Line 205: Add reference for Decontam’s prevalence method.

Reference added: Davis NM, Proctor D, Holmes SP, Relman DA, Callahan BJ (2017). “Simple statistical identification and removal of contaminant sequences in marker-gene and metagenomics data.” bioRxiv, 221499. doi: 10.1101/221499.

Line 210: >25,000 ASVs or a total of >25,000 ASV abundances?

Text changed from “but all remaining samples had >25,000 ASVs” to “but each remaining sample had >25,000 ASVs”

Line 223: Bray-Curtis dissimilarity

Change made.

Line 532: What does concentration of bacteria mean? Will it be abundance?

Response: “concentration” changed to “abundance”

---

## [Decision Letter · Decision Letter 1]

10 Jan 2022

Sodium molybdate does not inhibit sulfate-reducing bacteria but increases shell growth in the Pacific oyster *Magallana gigas*

PONE-D-21-29631R1

Dear Dr. Gold,

We’re pleased to inform you that your manuscript has been judged scientifically suitable for publication and will be formally accepted for publication once it meets all outstanding technical requirements.

Kind regards,

José A. Fernández Robledo, Ph.D.

Academic Editor

PLOS ONE

Additional Editor Comments (optional):

Thank you for addressing the reviewer's comments including mine.

Stay healthy and safe.

-j

Reviewers' comments:

Reviewer's Responses to Questions

**Comments to the Author**

1. If the authors have adequately addressed your comments raised in a previous round of review and you feel that this manuscript is now acceptable for publication, you may indicate that here to bypass the “Comments to the Author” section, enter your conflict of interest statement in the “Confidential to Editor” section, and submit your "Accept" recommendation.

Reviewer #1: All comments have been addressed

2. Is the manuscript technically sound, and do the data support the conclusions?

Reviewer #1: Yes

3. Has the statistical analysis been performed appropriately and rigorously? 

Reviewer #1: Yes

4. Have the authors made all data underlying the findings in their manuscript fully available?

Reviewer #1: Yes

5. Is the manuscript presented in an intelligible fashion and written in standard English?

Reviewer #1: Yes

6. Review Comments to the Author

Reviewer #1: (No Response)

7. PLOS authors have the option to publish the peer review history of their article (what does this mean?). If published, this will include your full peer review and any attached files.

Reviewer #1: No

---

## [Editor Report · Acceptance letter]

25 Jan 2022

PONE-D-21-29631R1 

Sodium molybdate does not inhibit sulfate-reducing bacteria but increases shell growth in the Pacific oyster *Magallana gigas*

Dear Dr. Gold:

I'm pleased to inform you that your manuscript has been deemed suitable for publication in PLOS ONE. Congratulations! Your manuscript is now with our production department. 

Kind regards, 

on behalf of

Dr. José A. Fernández Robledo 

Academic Editor

PLOS ONE